Beeswax waste improves the mycelial growth, fruiting body yield, and quality of oyster mushrooms (Pleurotus ostreatus)

Pan Chunlei 1
Sheng Chunge 1
Wang Kang 2
Zhang Yi 2
Liu Chunguang 1
Zhang Zhihao 1
Tao Liang 1
Lv Yang 1
Gao Fuchao 1 mdjgfc@126.com
1 Mudanjiang Branch, Heilongjiang Academy of Agricultural Sciences , Mudanjiang, Heilongjiang , China
2 College of Animal Science and Technology, Yangzhou University , Yangzhou, Jiangsu , China
Carter Dee
Electronic publication date: 2024 Dec 16
Publication date: 2024
Volume: 12
Electronic Location ID: e18726
Received 2024 Jul 31; Accepted 2024 Nov 26
Copyright: © 2024 Pan et al.
Copyright year: 2024
Copyright holder: Pan et al.
License: This is an open access article distributed under the terms of the Creative Commons Attribution License, which permits unrestricted use, distribution, reproduction and adaptation in any medium and for any purpose provided that it is properly attributed. For attribution, the original author(s), title, publication source (PeerJ) and either DOI or URL of the article must be cited.
License URL: https://creativecommons.org/licenses/by/4.0/

Keywords: Beeswax waste (BW), Pleurotus ostreatus, Yield, Nutritional quality

Funding: China Agriculture Research System CARS44 This work was supported by the China Agriculture Research System (No. CARS44). The funders had no role in study design, data collection and analysis, decision to publish, or preparation of the manuscript.

==============================
Heilongjiang Province has the third largest bee population in China, producing over 2,000 tons of beeswax waste (BW) each year. Most of this BW is discarded or burned. Therefore, we urgently need to find sustainable applications of BW. Pleurotus ostreatus mushrooms, commonly referred to as oyster mushrooms, are cultivated for both food and medicine. The substrate used to grow P. ostreatus mushrooms often contains wheat bran as a nitrogen source. The goal of this study was to explore the feasibility of substituting this wheat bran with BW to cultivate P. ostreatus mushrooms. Five treatments were established, with BW making up 0%, 3%, 5%, 7%, and 9% of the total substrate, and the effects on the mycelial growth and development, biological efficiency (BE), and yield were evaluated along with changes in the chemical biomass composition of the fruiting bodies. Adding BW increased the number of days needed for primordia initiation and the number of days between flushes of P. ostreatus mushrooms. With increasing BW, the total fresh weight of P. ostreatus mushrooms first increased and then decreased. The 5% BW treatment resulted in the highest yield and biological efficiency (BE) of 1,478.96 ± 9.61 g bag−1 and 92.43 ± 0.60%, respectively, which exceeded the values of the control by 4.14% (control: 1,420.15 ± 9.53 g bag−1 and 88.76 ± 0.60%, respectively). The 5% BW treatment also resulted in the highest mushroom crude protein content (23.47 ± 0.18 g 100 g−1), which was 28.18% higher compared with the control (18.31 ± 0.05 g 100 g−1). The 9% BW treatment resulted in the highest crude polysaccharide content (10.33 ± 0.76 g 100 g−1), which was 2.42-fold that of the control (4.26 ± 0.30 g 100 g−1). This study suggests that BW could serve as an effective source of nitrogen to cultivate P. ostreatus. BW is a promising, cost-effective, and efficient additive to mushroom substrate, improving the yield and quality of P. ostreatus mushrooms while providing a sustainable use for an otherwise difficult to dispose of waste product.

Introduction

China is one of the primary producers of honey worldwide. The number of bee-keepers and bee colonies and the amount of honey produced in China rank first in the world (Chen et al., 2012). The amount of honey produced in China was estimated to be 461,900 tons in 2022 and 463,500 tons in 2023 (National Bureau of Statistics, 2024; https://www.stats.gov.cn/) with the output increasing each year. Heilongjiang Province has the third largest bee population in the country (Wang, 2023). There were estimated to be more than 1.34 million bee colonies in Heilongjiang Province in 2023 (Li, 2023).

With the rapid development of the bee industry, more waste is being produced. This waste is difficult to dispose of and can negatively impact resources and the environment. In particular, beeswax waste (BW) is the residue left over after extracting beeswax from old honeycombs. The primary components of BW are bee carcasses, feces of bee larvae, residual honey, pollen and cocoons after insect rearing, deteriorated royal jelly, gum, and unpressed beeswax. One hundred pieces of old honeycomb can result in 20–30 kg of BW (Zhao & Zhao, 1991). Therefore, it is estimated that more than 2,000–3,000 tons of BW are produced each year in Heilongjiang Province.

BW is a solid material and is reported to be approximately 30% protein and lipid (Chen, 1979). It has a high potential for use in agricultural sectors to increase soil structure and fertility (Pourghasemian, Moradi & Iriti, 2023) as it is an organic amendment that is inexpensive, natural, eco-friendly, and easily available. However, there is little research on the use of BW in agriculture (Moradi, Pourghasemian & Naghizadeh, 2019). In China, BW is usually piled up on apiaries or burned as waste, which seriously pollutes the environment and wastes resources. This highlights the critical need for scientific analyses on sustainable uses of BW.

Pleurotus ostreatus is one of the most widely cultivated edible fungi in the world (Nam et al., 2018). Commonly known as oyster mushrooms, the fruiting bodies are favored by consumers for their high levels of nutrients, such as fiber, carbohydrates, fat-soluble vitamins and essential minerals, and low fat content (Kong et al., 2020; Abou Fayssal et al., 2021). In addition to their culinary appeal, P. ostreatus mushrooms comprise a group of edible fungi with important medicinal biotechnological and environmentally friendly applications (Elisashvili et al., 2008). P. ostreatus is easy to cultivate and can grow on various agricultural wastes; its short cultivation period and low production cost (Mahari et al., 2020) have led to a substantial demand in recent years and an increasingly broad market. As a result, P. ostreatus is the second most commonly produced variety of mushroom in the world (Grimm & Wösten, 2018; Kuforiji & Fasidi, 2009).

In recent years, cultivating Pleurotus spp. has become a promising method for converting lignocellulosic residues (plant-based waste materials) into protein-rich food. This approach utilizes renewable resources and contributes to food security by providing a sustainable source of nutrition (Naraian, Narayan & Srivastava, 2014).

In China, many regions cultivate P. ostreatus mushrooms as part of projects to eliminate poverty and promote rural revitalization, with the mushrooms playing an important role in ecological agriculture construction and rural economic development (Yang, 2019).

In the cultivation of P. ostreatus mushrooms, substances with high nitrogen contents, such as rice (Oryza sativa) bran and wheat (Triticum aestivum) bran, are often added to the primary growth substrate to promote the yield and quality of mushrooms (Moonmoon et al., 2011; Elisa et al., 2017). However, this also increases the production costs. Therefore, we need to find low-cost and efficient nitrogen sources to improve the quality and efficiency of P. ostreatus mushroom cultivation.

This study evaluated the effects of adding BW to the growth substrate on various parameters of P. ostreatus, including its mycelial growth and development, enzyme activities, yield of fruiting bodies, and composition of nutrients. The results indicate that BW, if applied at an appropriate percentage (5%), is a novel and sustainable nitrogen source for efficient cultivation of high-quality mushrooms. Moreover, the 9% BW treatment resulted in the highest crude polysaccharide content. To our knowledge, this is the first scientific study on the application of BW in the cultivation of edible fungi in China.

Materials and Methods

Fungal material

P. ostreatus samples were obtained from the Mudanjiang Branch of the Heilongjiang Academy of Agricultural Sciences (Mudanjiang, China). The strain was grown on potato dextrose agar (PDA) (200 g peeled, sliced and boiled potatoes, 20 g dextrose, and 20 g agar L−1) at 25 °C.

Substrate preparation

BW was obtained from the Mudanjiang Branch of the Heilongjiang Academy of Agricultural Sciences and the Raohe Northeast Black Bee Industry (Group) Co., Ltd. (Raohe, China). The other materials used to prepare the substrate were purchased locally.

Beeswax waste preparation

All the materials were dried in the sun. The dried BW was crushed with a Xin-zhuohui-FSJ grinder (Zhuohui Machinery Co., Ltd., Zhengzhou, China) and passed through an inner screen with 2-cm holes.

Analysis of the components of beeswax waste

The dry matter (DM) of the samples was determined after they had been dried to a constant weight at 65 °C. The contents of total carbon (TC) and total nitrogen (TN) were estimated as described by Dundar, Acay & Yildiz (2009).

Beeswax waste fermentation

BW was fermented as follows: The BW was placed in a pile, and the temperature of the pile was recorded. When the temperature of the pile exceeded 60 °C, it was held at this temperature for 24 h. Then, the pile was mixed and re-piled. In general, the pile required mixing 3–4 times during the fermentation period. The water content of the material was maintained at 55–65% (w/w). After the primary fermentation, Aspergillus oryzae was mixed with the BW at 0.15% (w/w) and the mixture was piled for a second round of fermentation. When the pile temperature was 32–40 °C, this temperature was maintained for 48–60 h. The pile was then re-mixed and re-piled. When the pile temperature increased to 55–65 °C, the temperature was maintained for 4–6 d. The pile was then cooled to room temperature (25–26 °C), completing the fermentation process.

Final substrate preparation

Corn cobs were used as the primary material in the growth substrate. First, corn cobs were chopped into small pieces (1–2 cm) with pruning shears, crushed, and passed through a 2-cm screen. Then, corn cobs were pre-wetted, and, after 12 h, wheat bran and lime were evenly spread on the surface of the pre-wetted material. The material was evenly mixed, piled, and then rows of vertical air holes were established in the pile. When the temperature at 20 cm below the maximum height exceeded 60 °C, the temperature was maintained for 24 h. The pile was then re-mixed and re-piled until it reached a temperature of 65 °C for approximately 2 h and then re-piled again. This continued until the pile had been re-mixed and re-piled 3–5 times. After fermentation was complete, the substrate was loose and elastic, not lumpy or sticky. At this stage, BW was added to the corn cob substrate so that the final ratios of each material in the growth substrate for each treatment followed those shown in Table 1. The moisture content in the final substrate was adjusted to 55–65% (w/w).

Table 1 Substrate composition for Pleurotus ostreatus cultivation.

Material	Percentage (DM) in each treatment	
CK	T1	T2	T3	T4	
Corncob	85	85	85	85	85	
Wheat bran	12	9	7	5	3	
BW	0	3	5	7	9	
Lime	3	3	3	3	3	
Note:

DM, dry matter; BW, beeswax waste; T, treatment; CK, control.

Mushroom cultivation

The prepared substrates were placed in 25 cm × 42 cm polyethylene bags that were 0.0015 cm thick with a packing density of 1,600 g of substrate (dry weight) per bag. The spawn was spread on the substrates to inoculate the cultivation bags using 20% (w/w) of substrate dry weight. In total there were 30 bags per replicate and three replicates per treatment. The inoculated bags were kept in the dark in the spawn running room at 25 °C and 60–65% relative humidity (RH). Then, bags were moved to a greenhouse at 17–22 °C and 80–90% RH and 100–200 Lux for primordia initiation and the fruiting body growth. The mycelial growth rate was determined as the height of the mycelia in the colonized culture bags (mm) divided by the incubation time (d). The time that it took the primordia to initiate (days from inoculation to the development of pinhead fruiting bodies) was noted. The time interval between flushes was considered to be the number of days between the first flush of mushrooms and the second flush.

Harvest and determination of the biological efficiency

Mushrooms were harvested when the mushroom caps were slightly rolled up at the margins as described by Yang, Guo & Wan (2013). The fruiting bodies were weighed after harvest, and the number of harvested bags was counted. These data were used to calculate the mushroom weights and biological efficiency (BE) at the end of the harvesting period. The yield was calculated as shown in Eq. (1):

(1) Mushroomweight(gbag−1)=Totalweightofharvestedfreshmushrooms/totalnumberofharvestedbags.

The BE was calculated as shown in Eq. (2):

(2) BE(%)=Weightoffreshmushroomsharvestedperbag/weightofdrysubstrateperbag×100.

Determination of enzyme activity

The substrate was sampled at 10, 20, and 30 d after inoculation, as well as during the formation of primordia and the maturation of fruiting bodies. Each sample size was 2 g. The sampling point was approximately 2 cm below the material surface. Three samples were randomly selected and quickly frozen in liquid nitrogen. Frozen samples were stored at −80 °C until further use. The substrates were thawed at 4 °C in an ice water bath. The substrates were then centrifuged at 8,000 × g or 10,000 × g for 10 min at 4 °C, and the supernatants that contained the enzymes were assayed.

Commercial test kits (Suzhou Comin Biotechnology Company, Ltd., Suzhou, China) were used to assay enzymes according to the manufacturer’s instructions. Briefly, laccase was assayed using the 2,2-azino-bis (3-ethylbenzthiazoline-6-sulphonic acid) (ABTS) method (Han et al., 2021). The amount of enzyme required to convert 1.0 μmol min−1 of ABTS was defined as one unit of activity. Carboxymethyl cellulase (CMCase) was assayed as described by Santa-Rose et al. (2018) using an enzyme-labeled instrument (RT-6100; Nidek, San Jose, CA, USA).

Fruiting body nutrient analysis

Fruiting bodies were randomly collected from each treatment group in the first flush, and the nutritional content was analyzed by Beijing Jinyan Innovation Technology Co., Ltd. (Beijing, China). The contents of crude polysaccharide (China Agricultural Press, 2023), crude fat (GB 5009.6, 2016), crude fiber (GB/T 5009.10, 2003), crude protein (GB 5009.5, 2016), and ash (GB 5009.4, 2016) were determined as described in the National Food Safety Standards.

Statistical analysis

One-way analysis of variance (ANOVA) and least significant difference (LSD) multiple range tests were used to determine differences in variables among the five treatments at a 95% confidence level (P < 0.05). All statistics were performed using SPSS 26.0 for Windows (IBM, Inc., Armonk, NY, USA). The normality of the data was tested using the Shapiro-Wilk method. Correlations were determined using the Pearson coefficient. Figures were initially plotted using GraphPad Prism 8 (San Diego, CA, USA) and then modified manually to improve presentation.

Results

Components of beeswax waste

The carbon-nitrogen ratio (C/N) of substrate is one of the key factors affecting the cultivation of edible fungi. The contents of total carbon (TC) and total nitrogen (TN) of BW were determined in this study. The TC content of the BW was 49.56%, while that of the TN was 1.86%. The C/N ratio of the BW was 26.65:1.

Mycelial growth and development

To assess the effect of BW on P. ostreatus mycelial growth and development, five levels of BW application—0% (CK), 3% (T1), 5% (T2), 7% (T3), and 9% (T4)—were investigated and the growth rate of mycelia as well as the primordial initiation time and the time interval between flushes were determined. The results showed that the addition of BW to P. ostreatus growth substrate affected the rate of mycelial growth, the time of primordial initiation, and the time interval between flushes (Table 2, Tables S1 and S2). The rate of mycelial growth in the different treatments followed the trend: T2 > T3 > T1 > CK > T4. The mycelia in T2 (5% BW) grew significantly faster than T4 mycelia (P < 0.05). However, there were no significant differences between the BW treatments and the control treatment. BW addition delayed primordial initiation. In T4 (9% BW), the primordial initiation time was 50.33 ± 1.53 d, which was significantly longer than that of other treatments (P < 0.05), including the control. BW also increased the time interval between flushes of P. ostreatus. T3 (7% BW) resulted in the longest time interval (13.67 ± 0.58 d), significantly longer than that of the control (11.67 ± 1.53 d) (P < 0.05).

Table 2 Growth and development of Pleurotus ostreatus mushrooms on different substrates.

Treatment	Mycelial growth rate
(mm d−1)	Primordial initiation time (days)	Time interval between flushes (days)	
CK	6.42 ± 0.45ab	43.33 ± 1.15c	11.67 ± 1.53b	
T1	6.43 ± 0.56ab	46.33 ± 1.15b	12.33 ± 0.58ab	
T2	6.89 ± 0.56a	47.33 ± 1.53b	12.67 ± 1.53ab	
T3	6.69 ± 0.58ab	47.00 ± 1.73b	13.67 ± 0.58a	
T4	6.34 ± 0.52b	50.33 ± 1.53a	13.33 ± 0.68ab	
Notes:

Means ± SD are shown. SD, standard deviation.

Different lowercase letters (a, b, c) indicate significant differences (α = 0.05, ANOVA, LSD test).

T, treatment; BW, beeswax waste; CK, control.

CK: 85% corncob, 12% wheat bran, 3% lime; T1: 85% corncob, 9% wheat bran, 3% BW, 3% lime; T2: 85% corncob, 7% wheat bran, 5% BW, 3% lime; T3: 85% corncob, 5% wheat bran, 7% BW, 3% lime; T4: 85% corncob, 3% wheat bran, 9% BW, 3% lime.

Yield and biological efficiency

The fresh weight of each flush, BE, and total yield of P. ostreatus are shown in Table 3 and Table S4. The mushrooms in the first flush of T1 and T2 weighed 674.19 ± 14.30 g bag−1 and 678.16 ± 4.03 g bag−1, respectively, and these values did not differ significantly from those of the control (681.42 ± 5.08 g bag−1). However, T3 and T4 resulted in significantly lower mushroom weights (651.97 ± 12.91 g bag–1 and 649.07 ± 9.97 g bag−1, respectively) than the other BW treatments and the control treatment (P < 0.05).

Table 3 Fresh weight and biological efficiency of the Pleurotus ostreatus mushrooms grown on different substrates (mean ± SD).

Treatment	Fresh weight of the mushrooms (g bag−1)	BE (%)	
First flush	Second flush	Third flush	Total	
CK	681.42 ± 5.08a	473.53 ± 9.77ab	265.21 ± 7.11b	1420.15 ± 9.53b	88.76 ± 0.60b	
T1	674.19 ± 14.30a	475.17 ± 9.51ab	277.50 ± 7.38b	1426.85 ± 10.19b	89.18 ± 0.64b	
T2	678.16 ± 4.03a	482.05 ± 4.58a	318.75 ± 9.42a	1478.96 ± 9.61a	92.43 ± 0.60a	
T3	651.97 ± 12.91b	465.47 ± 6.21bc	281.09 ± 10.91b	1398.53 ± 7.50c	87.41 ± 0.47c	
T4	649.07 ± 9.97b	457.26 ± 11.25c	240.48 ± 8.55c	1346.81 ± 5.36d	84.18 ± 0.34d	
Notes:

Different lowercase letters (a, b, c) indicate significant differences (α = 0.05, ANOVA, LSD test).

SD, standard deviation; T, treatment; BE, biological efficiency; BW, beeswax waste.

CK: 85% corncob, 12% wheat bran, 3% lime; T1: 85% corncob, 9% wheat bran, 3% BW, 3% lime; T2: 85% corncob, 7% wheat bran, 5% BW, 3% lime; T3: 85% corncob, 5% wheat bran, 7% BW, 3% lime; T4: 85% corncob, 3% wheat bran, 9% BW, 3% lime.

The fresh weight of the second flush ranged from 457.26 ± 11.25 to 482.05 ± 4.58 g bag−1 and followed the trend: T2 > T1 > CK > T3 > T4. There were no significant differences in yield between T1, T2, or T3 and the control. However, T4 (457.26 ± 11.25 g bag−1) resulted in a significantly lower yield compared with the control treatment (473.53 ± 9.77 g bag−1).

For the third flush, the mushroom fresh weights for T1, T2, T3 and T4 were 277.50 ± 7.38, 318.75 ± 9.42, 281.09 ± 10.91, and 240.48 ± 8.55 g bag−1, respectively. However, the fresh weights of T1 and T3 did not differ significantly nor were they significantly different from the fresh weight of control (265.21 ± 7.11 g bag−1). The mushroom yield of T2 was significantly higher than yields of the other treatments, and T4 had a significantly lower yield than yields of the other treatments (P < 0.05).

BW substantially affected total P. ostreatus mushroom yield. The total yields of CK and T1 were similar (1,420.15 ± 9.53 and 1,426.85 ± 10.19 g bag−1, respectively). The total yield and BE of T2 were 1,478.96 ± 9.61 g bag−1 and 92.43 ± 0.60%, respectively, which was significantly higher than those of the other treatments (P < 0.05). The total yields of T3 and T4 were 1,398.53 ± 7.50 and 1,346.81 ± 5.36 g bag−1, respectively. These two treatments had significantly different total yields from each other and significantly lower yields than those of the other treatments (P < 0.05).

Enzyme activities

Laccase and carboxymethyl cellulase (CMCase) are the main enzymes secreted by P. ostreatus mycelia. These enzymes degrade the lignocellulose in the growth substrate. The nutritional content of the growth substrate can affect mycelial cells and therefore extracellular enzyme activity. Laccase activity fluctuated considerably with sampling time in all treatments (Fig. 1, Tables S3 and S6). The laccase activities for all treatments were determined at 10 d after inoculation and ranged from 234.32 ± 15.06 to 285.92 ± 9.88 U L−1. The laccase activity did not differ significantly between T1 and T2, but these treatments had significantly higher laccase activities than other treatments (P < 0.05). There were no significant differences in laccase activity at 20 d after inoculation between T1, T2, or T3 and T4. However, T3 resulted in significantly higher laccase activity compared with the control treatment (P < 0.05). The laccase activity at fruiting maturity in T4 was significantly lower than that in the control group (P < 0.05). However, there were no significant differences between the other BW treatments and the control treatment. The highest laccase activities for all treatments were observed at 30 d after inoculation and ranged from 310.32 ± 14.90 to 389.85 ± 26.21 U L−1. The laccase activity then decreased sharply, and the lowest values for all treatments were observed during the primordial initiation period (146.38 ± 11.23 to 206.80 ± 16.22 U L−1). Both at 30 d after inoculation and during the primordial initiation period, T2 had the highest laccase activities, with activities significantly higher than those of the control group (P < 0.05).

Figure 1 Radar maps of enzyme activities (units: U L−1) at 10, 20, and 30 d after inoculation, the primordial initiation period, and the fruiting maturity period for each treatment.

(A) Laccase activity; (B) carboxymethyl cellulase activity. BW, beeswax waste; CK, control; T, treatment. CK: 85% corncob, 12% wheat bran, 3% lime; T1: 85% corncob, 9% wheat bran, 3% BW, 3% lime; T2: 85% corncob, 7% wheat bran, 5% BW, 3% lime; T3: 85% corncob, 5% wheat bran, 7% BW, 3% lime; T4: 85% corncob, 3% wheat bran, 9% BW, 3% lime.

Similar to laccase, the lowest carboxymethyl cellulase (CMCase) activities were observed during the primordial initiation period (135.05 ± 25.98 to 216.77 ± 16.18 U L−1). Conversely, the highest CMCase activities were observed during the fruiting maturity period (438.30 ± 26.16 to 516.79 ± 26.48 U L−1). While there were no significant differences in CMCase activity among the treatments on day 10 after inoculation, T2 had significantly higher CMCase activity than that of the control on day 20 after inoculation as well as in the primordial initiation and fruiting maturity periods (P < 0.05). However, the CMCase activity of T2 did not differ significantly from than that of the control on day 30 after inoculation.

Nutritional value

The nutrient content of edible fungi, including the higher crude polysaccharide, higher crude protein, lower crude fat, proper crude fiber and ash content, are important for its value as a food. In this study, nutrient contents in the P. ostreatus fruiting bodies differed significantly among the different treatments (Fig. 2 and Table S5). The CK mushrooms had the highest ash content (7.10 ± 0.06 g 100 g−1) and the lowest crude polysaccharide (4.26 ± 0.30 g 100 g−1) and crude fiber contents (4.86 ± 0.05 g 100 g−1). The T1 mushrooms had the highest crude fat content (0.72 ± 0.03 g 100 g−1) and the lowest ash content (6.81 ± 0.13 g 100 g−1). The T2 mushrooms had the highest crude protein content (23.47 ± 0.18 g 100 g−1). The T3 mushrooms had the highest crude fiber content (6.82 ± 0.04 g 100 g−1) and the lowest crude fat content (0.56 ± 0.04 g 100 g−1). The T4 mushrooms had the highest crude polysaccharide content (10.33 ± 0.76 g 100 g−1) and the lowest crude protein content (15.89 ± 0.20 g 100 g−1).

Figure 2 Nutritional components of Pleurotus ostreatus mushrooms grown on different substrates.

Different lowercase letters (a, b, c) above bars indicate significant differences (α = 0.05, ANOVA, LSD test). BW, beeswax waste; CK, control; T, treatment. CK: 85% corncob, 12% wheat bran, 3% lime; T1: 85% corncob, 9% wheat bran, 3% BW, 3% lime; T2: 85% corncob, 7% wheat bran, 5% BW, 3% lime; T3: 85% corncob, 5% wheat bran, 7% BW, 3% lime; T4: 85% corncob, 3% wheat bran, 9% BW, 3% lime.

With increasing BW, the crude polysaccharide content increased and the crude protein content first increased and then decreased. There were no clear patterns in crude fat content. The addition of BW significantly increased the crude fiber content of the P. ostreatus mushrooms. Adding BW decreased ash content compared with the control, but the ash content increased with increasing BW addition.

Correlation analysis

Mycelial growth and fruiting body formation are the two primary stages of edible fungus cultivation. The mycelial growth rate, primordium initiation time, and time interval between flushes are three important indicators of mycelial growth and development. In this study, the relationship between mycelial growth and fruiting body yield (fresh weight) of P. ostreatus was determined (Fig. 3). The mycelial growth rate was significantly positively correlated with the total fresh weight of P. ostreatus fruiting body (P < 0.05), while neither the primordium initiation time nor time interval between flushes were significantly correlated with the total yield of P. ostreatus fruiting body.

Figure 3 Correlations among agronomic characteristics and total fresh weight of Pleurotus ostreatus mushrooms.

Values are Pearson’s correlation coefficients. Green and brown indicate the degree of association. Brown, positive correlation. Green, negative correlation. *p < 0.05.

Discussion

This study aimed to explore the feasibility of substituting wheat bran with BW as a nitrogen source in P. ostreatus cultivation. The key findings indicate that BW addition at different weight ratios had different effects on the growth and development of P. ostreatus mycelia, enzyme activity, the total fresh weight and BE, and the nutrient content of P. ostreatus mushroom.

The cultivation substrate can influence P. ostreatus mushroom growth (Das et al., 2014; Sher, Al-Yemeni & Khan, 2011). In the current study, the number of days required for primordial initiation varied from 43.33 ± 1.15 d to 50.33 ± 1.53 d, which is within the range reported by Zakil et al. (2021). In their article, the lowest growth period (29 ± 3.00 d) for pin head formation of P. ostreatus was obtained in 100% rubber tree sawdust and a long duration (131 ± 20.07 d) for primordial development was recorded in 100% oil palm empty fruit bunch. Baysal et al. (2003) reported that P. ostreatus grown on waste paper took 2–3 weeks for fruit body formation after spawn running. Our results agree with their findings. The number of days between the first flush and the second flush in our study ranged from 11.67 ± 1.53 d to 13.33 ± 0.68 d, which was longer than the 6–10 d reported by Mandeel, Al-Laith & Mohamed (2005).

In our study, the mycelial growth rate was positively correlated with mushroom yield (total fresh weight). This result is similar to that of Pleurotus florida cultivated on corncob substrate (Naraian et al., 2008), but in contrast with P. ostreatus cultivated on different lignocellulosic by-products (Obodai, Cleland-Okine & Vowotor, 2003) and Lepista sordida cultivated on grain crop residues (Sheng et al., 2024), in these studies the mycelial growth rate did not correlate with yield or BE. This may be because the mycelial growth and development of P. ostreatus is not only related to the cultivation substrate but also to the genotype, cultivation conditions, and other factors (Shen et al., 2021; Zhang & Yang, 2017; Ejigu et al., 2022). In this study, laccase activity peaked at day 30 after inoculation, which corresponded with the time that it took the mycelium to fill the cultivation bag. This indicates that the mycelium was active at this time and accumulating energy for the initiation of primordia. Then, during primordial initiation, laccase activity was lower, likely because of the change in the physical structure of the mycelium during this time. For CMCase, the lowest activities were observed during primordial initiation, while activities peaked during the period of fruiting body maturity. These results agree with those of Shen et al. (2010). Interestingly, the activities of both enzymes were significantly higher in T2 than in the control at the points when enzyme activities were highest and lowest. T2 also resulted in a higher yield and BE, indicating that this treatment provided a good nutrient supply to mycelial cells, resulting in rapid growth of hyphae and high nutrient contents in the fruiting bodies.

Nitrogen is an important nutrient for the growth and development of edible mushrooms (Moonmoon et al., 2011). Different fungi strains respond differently to nitrogen sources. Within a certain range, increasing the nitrogen content can increase the yield of fruiting bodies (Wen et al., 2018). Our study found that with increasing BW supplementation, the total yield and BE of P. ostreatus tended to first increase and then decrease. This result is similar to those of Guo et al. (2024) and Hu et al. (2022), who reported an initial increase and then decrease in the yield of mushrooms as more nitrogen was added. A lack of nitrogen in the substrate can lead to an imbalance of nutrients in the fungus and thus is not conducive to high yields. According to nitrogen utilization by fungi, nitrogen sources can be divided into primary and secondary nitrogen sources (Marzluf, 1997). In this study, T2 had a significantly higher total yield than the control. However, the mushroom yields of the first and second flushes from T2 did not differ significantly from those of the control, while the third flush of mushrooms had significantly higher yields than the control. We speculated that wheat bran was used as a primary nitrogen source and BW was a secondary nitrogen source. This indicates that BW is a slow-release and long-acting source of nitrogen. Moreover, there was evidence that BW stimulated the secretion of extracellular laccase and CMCase by P. ostreatus mycelia, which likely accelerated the degradation of nutrients, such as cellulose and lignin in the substrate. This could explain the increased mycelial growth, mushroom yields, and accumulation of nutrients in mushrooms. The potential mechanism by which BW acts as a supplemental nitrogen source merits further study. In our study, the first mushroom flush produced approximately 50% of the total yield, while the second flush produced 30% of the total yield. This was consistent with the cultivation of P. ostreatus on Hypsizigus marmoreus mushroom substrates (Wang et al., 2015). The BE and yields of the first flush were consistently higher than those of the second flush across all treatments. These results are similar to those of a study in which Agrocybe cylindracea and P. ostreatus were cultivated on agricultural and forestry byproducts (Koutrotsios et al., 2014). In contrast, when Auricularia polytricha was cultivated on the sawdust waste of spent mushrooms (Wu, Liang & Liang, 2020), the second flush had higher yields and BE than the first flush.

The nutritional composition of mushrooms varies among different species and strains (Hoa, Wang & Wang, 2015), and is substantially affected by the environmental conditions and cultivation substrate (Xu et al., 2016; Gupta et al., 2013). In the current study, the substrate composition significantly affected the nutritional content of the mushrooms. The average crude protein content of P. ostreatus mushrooms in T2 was 23.47%, which was within the crude protein range reported in other articles (7.02% to 41.6%, Table 4). The average crude fat content in this study was 0.60%, which is low compared with values from other studies (0.50–5.45%). The ash concentration of oyster mushrooms ranges from 5.49% to 9.80%, while the value of P. ostreatus in this study was 6.88%. However, the average crude polysaccharide content of P. ostreatus in this study was 6.77%, which was higher than that of P. ostreatus mushrooms grown on a substrate containing common reeds (Phragmites australis). The average crude fiber content in this study was 5.27%, which is similar to the content reported by Ulziijargal & Mau (2011) (5.33%) but lower than that reported by Boadu et al. (2023) (9.09%). In summary, the current study demonstrates that mushrooms grown on BW are suitable for the health and nutritional needs of modern consumers.

Table 4 Nutritional content of Pleurotus ostreatus mushrooms from previous studies and this article.

Crude protein (%)	Crude fat (%)	Ash (%)	Crude polysaccharide (%)	Crude fiber (%)	Reference	
19.08	1.60	5.49	5.00–6.00		Li et al. (2023)	
7.02	1.40	5.72			Reis et al. (2012)	
23.85	2.16	7.59		5.33	Ulziijargal & Mau (2011)	
41.60	0.50	6.00			Kirbağ & Akÿuz (2010)	
16.69	5.45	6.70			Jaworska, Bernaś & Mickowska (2011)	
18.36	4.25	9.80		9.09	Boadu et al. (2023)	
23.47	0.60	6.88	6.77	5.27	This article (T2)	

This article reports the first attempt at using BW for edible mushroom cultivation. We confirmed the feasibility and potential of BW for P. ostreatus mushroom cultivation. However, the experimental design has certain limitations, as it lacks a treatment group in which 100% of the wheat bran was replaced by BW, and therefore the results are not comprehensive. Despite this, the current study showed that the total yield was significantly reduced when the amount of BW was more than 7% (7% and 9% BW resulted in yields of 1,398.53 ± 7.50 and 1,346.81 ± 5.36 g bag−1, respectively). In future research, a complete replacement treatment group should be included to optimize the experimental design. In addition, the effects of BW on mushroom flavor, texture, and fruiting body morphology, as well as the mechanisms by which BW affects the growth and development of P. ostreatus, need further research.

Before BW can be widely used in edible mushroom cultivation, the following three issues need to be addressed. First, the safety and stability of BW needs to be ensured, and effective methods need to be developed to remove impurities and potentially harmful substances such as pesticide residues and heavy metals in BW. Second, the fermentation efficiency of BW and the quality of fermentation products need to be improved. The quality of fermentation materials plays an important role in the growth and development of mushrooms (Kong et al., 2020). In this study, secondary fermentation of BW was used to improve the quality of fermentation products using Aspergillus oryzae (Ruan, 2023). However, the BW fermentation process needs to be further optimized (such as fermentation time, temperature, humidity, ventilation, etc.) to improve the utilization rate and nutritional value of BW. Third, further research is needed on the adaptability of other edible mushroom varieties to BW fermentation products, as well as the feasibility and economic benefits of BW in mushroom production.

Conclusions

The effects of BW on the growth of mycelia, BE, yield, and nutritional value of P. ostreatus were evaluated. This study found that BW improved the BE, total yield, and nutritional value of P. ostreatus by affecting its accumulation of proteins and polysaccharides. Therefore, BW is a suitable nitrogen source for the cultivation of P. ostreatus. This provides a practical and sustainable use for BW, reducing the environmental impact of waste accumulation.

Supplemental Information

Supplemental Information 1 Supplementary material.

We wish to thank Wei Jiang and Haiyang Jiang from Raohe Northeast Black Bee Industry (Group) Co., Ltd. for their selfless assistance in providing the raw material of beeswax waste.

Additional Information and Declarations

Competing Interests

Author Contributions

Data Availability

The authors declare that they have no competing interests.

Chunlei Pan conceived and designed the experiments, performed the experiments, prepared figures and/or tables, and approved the final draft.

Chunge Sheng conceived and designed the experiments, prepared figures and/or tables, and approved the final draft.

Kang Wang analyzed the data, authored or reviewed drafts of the article, and approved the final draft.

Yi Zhang analyzed the data, authored or reviewed drafts of the article, and approved the final draft.

Chunguang Liu performed the experiments, prepared figures and/or tables, and approved the final draft.

Zhihao Zhang performed the experiments, prepared figures and/or tables, and approved the final draft.

Liang Tao analyzed the data, prepared figures and/or tables, and approved the final draft.

Yang Lv performed the experiments, prepared figures and/or tables, and approved the final draft.

Fuchao Gao conceived and designed the experiments, authored or reviewed drafts of the article, and approved the final draft.

The following information was supplied regarding data availability:

The raw measurements are available in the Supplemental File.

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
