# Peer review of "Beeswax waste improves the mycelial growth, fruiting body yield, and quality of oyster mushrooms (Pleurotus ostreatus)"

_PeerJ, doi:10.7717/peerj.18726_

## Round 0.1 · original submission · Major Revisions

You have received two comprehensive expert reviews of your manuscript, which outline important changes that need to be considered before the manuscript is suitable for publication. Please revise the manuscript in line with these reviews. In particular, ensure that the results are discussed in a manner that is consistent with your findings and are not overstated, and include any limitations. Reviewer #1 suggests a substantial re-write of the discussion to improve the structure and flow. Both reviewers note problems in the numbering and inclusion of figures and tables in the text and this must be corrected. Both also note a lack of contextualisation and consideration of the broader implications of the study. Please address all of the reviewer comments and questions in your point-by-point rebuttal letter.

·

Basic reporting

The introduction provides a good background, but the link between BW waste and its potential use as a nitrogen source in mushroom cultivation could be elaborated further. Is there any literature on nitrogen content of BW?

The manuscript follows a standard structure and includes relevant figures and tables. However, the figures both in the main text and supplementary material are incorrectly numbered. Figures 1 and 2 in the manuscript should be Figure 1 A and B, making all subsequent figures misnumbered. Table S3 and S5 are referred to in the text using incorrect numbers and Tables S1, 2, and 4 are never mentioned. Please ensure all figures and tables in both the main and supplementary files are correctly numbered and referred to in the text in the correct order.

Experimental design

While the methods are generally well described, there are some areas that need clarification:

Line 109-110 – how was water content measured and adjusted?

Line 112 – “at a rate of 0.15%” what does this mean? Is it concentration of 0.15% by weight? Please clarify.

Line 158 – how were they homogenized?

Please also describe the environmental conditions under which the mushrooms were cultivated, such as temperature, humidity, and light conditions. This information is essential for the reproducibility of the study.

Validity of the findings

The data provided appears robust, but the presentation of results could be clearer. Please include p-values for all mentions of statistical significance throughout the manuscript and correct ambiguities in figure legends as described below.

The discussion lacks a clear structure, with findings presented in a somewhat disorganised manner. To improve, the discussion should be broken into focused sections and systematically link findings to the research objectives and existing literature. Please see further comments below.

Additional comments

Introduction:

Lines 42, 44, 46 require proper references whether this be a website, report, book, etc.

Line 53 – reference?

Line 57 – says both “readily available” and “easily available”, please remove one of these

Line 60 – please change “bee farms” to “apiaries”

Line 67 – change to “In addition to their culinary appeal, …”

Line 71 – change “Therefore” to “As a result,”

Line 72 – “In recent years…” This is a confusing and complex sentence. Make this a new paragraph and rewrite for clarity. For example, “In recent years, cultivating Pleurotus spp. has become a promising method for converting lignocellulosic residues (plant-based waste materials) into protein-rich food. This approach utilizes renewable resources and contributes to food security by providing a sustainable source of nutrition.”

Methods:

Line 174 – write Least Significant Difference in full

Results:

**throughout entire results please include p-values for all mentions of significance

Line 181 – please add an introductory sentence at the beginning of this paragraph explaining what was done and why. For example, “In order to assess the effect of BW on P. ostreatus growth, treatments of 0% (T1), 3% (T2), 5% (T3), etc..”

Line 182 – add % BW to each treatment and explain what CK means

Line 226 – add context for why you chose to look at these particular enzymes and why they are important.

Line 227 – this is table S3 in the supplementary, please correct the ordering or numbering

Line 230 – what about 10 d, 20 d, and fruiting maturity period?

Line 237 – what about 30 d?

Line 239-240 – “These higher enzyme activities…” is too speculative for results, move to discussion

Line 247 – add context for why you chose to look at these particular nutrients and why they are important. Please indicate which are desirable or non-desirable traits.

Line 248 – this is table S5 in the supplementary, please correct the ordering or numbering

Discussion:

Line 273 – the explanation and results for Figure 3 (Figure 4 in the manuscript) need to be in the results section, not the discussion section. Please move this and provide a full description of the results.

In general, the discussion would benefit from a more logical structure and focused subsections. The discussion immediately currently jumps into random findings without first restating the objectives or framing the key discoveries. Please start with a brief recap of the study’s objectives and summarize the key findings. For example, “This study aimed to explore the feasibility of substituting wheat bran with beeswax waste (BW) in Pleurotus ostreatus cultivation. The key findings indicate that…”

The interpretation of the results is scattered. Discussion of each finding is not always tied back to the hypothesis or the relevant literature. Related studies are mentioned but not in a systematic way with some comparisons feeling sporadic rather than integrated. It could be improved by breaking down the interpretation of the results into distinct subheadings, for example: yield and biological efficiency, nutritional composition, enzyme activity and mushroom development.

The broader implications of the study are underdeveloped. Please add discussion of the environmental and practical implications of using BW on a larger scale. For example, are there any potential drawbacks, such as variability in BW composition or challenges in sourcing BW consistently?

Limitations are not addressed, leaving out important considerations for future research. Please add discussion of any limitations in study design or methodology that might impact interpretation of the results. For example, were there any limitations related to the consistency of BW as a substrate, or were there challenges in controlling the fermentation process?

Please suggest areas for future research, such as testing BW with other mushroom species or exploring its effects on mushroom flavour or texture.

The conclusion sentence “the results indicate that BW is a novel and sustainable nitrogen source for efficient cultivation of high-quality mushrooms” is overstated and does not entirely align with the overall findings of the work which found some concentrations of BW beneficial in certain parameters and others detrimental in certain parameters. A more nuanced conclusion is required.

Tables & Figures:

Please correct all figure numbers and check in the text.

Tables S1, S2, and S4 are never mentioned in the text.

Table 2 and 3 – these tables are not very big so please add %BW here for ease of understanding

Table 2 – what does the a, b, c mean at the end of the values?

Table 3 – legend states different lowercase letters indicate significant differences (p<0.05)” but does not explain what each letter means. Please explain what p value corresponds to what letter.

Figure 1 and 2 – these have been entered as two figures but are clearly parts (a) and (b) of the same figure. Please correct this.

Figure 3 – as above, what do the letters a, b, c, mean?

Figure 4 – was the normality of the data assessed before choosing Pearson correlations? Please clarify and add this to the method section. In addition, the legend says ** corresponds to (P<0.01) but there are no ** designations on the graph? Please delete if unnecessary or double check correlations were not missed on the graph.

Reviewer 2 ·

Basic reporting

The journal article written by Pan et al., reports using beeswax waste as a nitrogen source in Pleurotus ostreatus (oyster mushroom) production. I believe this article fits within the scope of the PeerJ journal.

Overall, the language in this article was sound, however, there were some issues with the flow of the article. For example, in line 47, remove the word “However” or join this paragraph with the previous section.

In lines 274-278 of the submitted manuscript, the structure of this paragraph should be flipped. As it is currently read, the findings of this article state that mycelial growth rate should not correlate to yield and biological efficiency.

Experimental design

In lines 101-110 of the submitted manuscript, I suggest combining the beeswax fermentation process as one whole section. There is also no mention of why Aspergillus oryzae was used in the fermentation process for beeswax waste, nor why the fermentation process was important for substrate preparation.

I suggest the authors measure their beeswax waste's total protein and lipid content.

I suggest the authors also try substituting 100% of the wheat bran with the beeswax waste or provide justification if not possible.

Validity of the findings

The authors may want to consider discussing and comparing the results between treatments to provide a more comprehensive analysis. Additionally, focusing the discussion on comparisons with other studies that focus on Pleurotus ostreatus would help better contextualize the findings of this study.

It would be beneficial for the authors to provide further clarification on the practicality of using beeswax waste in Oyster mushroom cultivation. Considering the substantial annual production of beeswax waste, it would be valuable for the authors to discuss potential strategies for scaling up the use of beeswax waste in Oyster mushroom cultivation to make it a more practical and impactful method of waste utilization.

Additional comments

Line 191: the table legend does not describe the lowercase letters in the table

In the reference list, please italicise all genus species names

In line 227 of the submitted manuscript, the authors refer to Table S1 of the supplementary materials, however, this should be corrected to Table S3

Line 248: Table S2 should be corrected to Table S5

The colours used for Figure 3 are misleading. Please consider using a different colour palette

---

## Round 0.2 · Minor Revisions

Thank you for resubmitting your revised manuscript and for taking on board the reviewers' comments. Both reviewers feel that the manuscript is now greatly improved. They have a few minor things that should still be addressed prior to acceptance.

·

Basic reporting

The authors have significantly improved their manuscript and satisfactorily responded to the majority of reviewer comments. However, a few issues remain:

All p-values in the manuscript are reported as “P<0.05”. Please report exact p-values (unless they are <0.001).

The descriptions of what ‘a, b, c, etc’ mean on Table 2 and Figure 2 are still unclear. The text now states “Different lowercase letters (a, b, c) above bars indicate significant differences (α = 0.05, ANOVA, LSD test).” Please indicate what each individual letter corresponds to both in terms of significance level and what is being compared.

Experimental design

Satisfactory

Validity of the findings

Satisfactory

Additional comments

Satisfactory

Reviewer 2 ·

Basic reporting

The resubmitted manuscript by Pan et al. reports the use of beeswax waste as a novel and sustainable nitrogen source for Pleurotus ostreatus cultivation. The rebuttal response and revised manuscript addressed many of the concerns raised in this manuscript's initial submission.
Overall, the language of this article was sound. Although there were some small grammatical issues that affected the flow of the article, I believe this article fits within the scope of this journal.

Lines 130-131: please change "water content" to "moisture content"
Line 150: Please correct "The data" to "This data was" of "These data were"
In Table S3: please change "Enzyme activities" to "Laccase activity"

Experimental design

no comment

Validity of the findings

no comment

Additional comments

For future research, I'd like to advise the authors to consider the forms of nitrogen (NH4+, NO3-, etc.) in bees waxwaste that are available for uptake by the mushrooms.
It was a pleasure to reread this article.

---

## Round 0.3 · Minor Revisions

Reviewer #1 requested that the designations (a, b, c) on figure 2 and table 2 be clarified. I am assuming that each letter indicates that the value is significantly different to the value designated by the other letters, but this needs to be clearly stated. Also, on Figure 2 you have a, b, c, d, and e. Please also check for Ash on Fig 2 where this is no b alone and I assume that there should be.

---

## Round 0.4 · accepted · Accept

Your article is now Accepted